# Endothelial Glycocalyx Preservation—Impact of Nutrition and Lifestyle

**DOI:** 10.3390/nu15112573

**Published:** 2023-05-31

**Authors:** Paula Franceković, Lasse Gliemann

**Affiliations:** Department of Nutrition, Exercise and Sports, University of Copenhagen, Universitetsparken 13, DK-2100 Copenhagen, Denmark

**Keywords:** endothelial glycocalyx, lifestyle diseases, cardiovascular health, obesity, diabetes, hypertension, nutrition therapy, Mediterranean diet, vitamin D, dietary sulfur, dietary nitrates, mechanotransduction, endotoxemia, intermittent fasting

## Abstract

The endothelial glycocalyx (eGC) is a dynamic hair-like layer expressed on the apical surface of endothelial cells throughout the vascular system. This layer serves as an endothelial cell gatekeeper by controlling the permeability and adhesion properties of endothelial cells, as well as by controlling vascular resistance through the mediation of vasodilation. Pathogenic destruction of the eGC could be linked to impaired vascular function, as well as several acute and chronic cardiovascular conditions. Defining the precise functions and mechanisms of the eGC is perhaps the limiting factor of the missing link in finding novel treatments for lifestyle-related diseases such as atherosclerosis, type 2 diabetes, hypertension, and metabolic syndrome. However, the relationship between diet, lifestyle, and the preservation of the eGC is an unexplored territory. This article provides an overview of the eGC’s importance for health and disease and describes perspectives of nutritional therapy for the prevention of the eGC’s pathogenic destruction. It is concluded that vitamin D and omega-3 fatty acid supplementation, as well as healthy dietary patterns such as the Mediterranean diet and the time management of eating, might show promise for preserving eGC health and, thus, the health of the cardiovascular system.

## 1. Introduction

The endothelial glycocalyx (eGC), sometimes referred to as the endothelial surface layer, is a protective layer of glycoconjugates (e.g., carbohydrates covalently linked to other molecules such as amino acids, proteins, lipids, etc.) that covers the luminal side of the endothelial cells (see Figure 1). In homeostatic conditions, this layer serves as an endothelial cell gatekeeper by controlling the permeability of substances from the blood to the interstitium and initiating the adhesion of blood-carried molecules to the cell surface [1]. In addition, the eGC is crucial for proper endothelial nitric oxide production and vasodilatation. Thus, the eGC serves an important regulatory function for the cardiovascular system [2]. Pathogenic destruction of the eGC is detected in a large number of cardiovascular conditions, including atherosclerosis [3,4], hypertension [5], diabetes mellitus [6], chronic kidney disease [7], ischemia reperfusion syndrome [8], and septic shock [9]. Most of the current research regarding the eGC is aimed towards acute care surgery and perioperative care, because derangement of the eGC is proposed to increase the severity of sepsis. Presently, little is known about the regeneration or prevention of its pathogenic destruction in randomized controlled clinical trials [10,11]. It is suggested that the main reason behind this is a lack of reliable detection techniques is due to the ex vivo instability of the eGC [12]. In addition, this important structure of the cardiovascular system has been overseen by researchers, funding bodies, and pharmaceutical companies [13]. Nevertheless, newly developed methods, such as side-stream dark-field imaging, orthogonal polarization spectral imaging, and improved fixation techniques, have shed new light on the eGC and its role in health and disease [14,15]. The in-depth eGC knowledge is a missing link in primary, secondary, and tertiary cardiovascular disease prevention and treatment. The eGC is, therefore, a novel target for various healthcare professionals such as nutritionists, dietitians, clinicians, surgeons, oncologists, researchers, and many others. The relationship between diet and eGC destruction and/or restoration is still an unexplored territory, but it might be particularly important in cardiovascular disease prevention and treatment. The aims of the present review are the following: (1) to provide an overview of current eGC knowledge—including factors that influence eGC structure and function in microcirculation—in healthy conditions and in chronic vasculature-related inflammatory pathologies; and (2) to describe perspectives of nutritional therapy, as well as diet- and lifestyle-related behaviors for eGC destruction prevention and regeneration.

## 2. The eGC in Healthy Conditions—Extended State of eGC

The eGC is a dynamic hair-like layer expressed on the apical surface of the endothelial cells, facing the lumen, throughout the vascular system. It is described as a “hair-like layer” because the structures of eGC resemble microscopic hair when observed under the microscope [16] (Figure 2). Additionally, the eGC is described as a “dynamic layer” because it undergoes continuous metabolic turnover that is dependent on the local environment; as such, in healthy, homeostatic conditions, the eGC is in a state of dynamic equilibrium with plasma proteins and exists in an ‘extended state’. Most commonly, the components consist of a proteoglycan backbone with many glycosaminoglycan attachment sites. Syndecans, glypican-1, biglycans, and perlecans are the typical proteoglycans of the glycocalyx. These proteoglycans mainly bind to heparan sulfate, which is the most abundant glycosaminoglycan of the eGC (50–90% of the total glycosaminoglycan pool). Other glycosaminoglycans that contribute to eGC integrity and permeability control are chondroitin, dermatan, and keratan sulfates [17,18]. The final integral part of the eGC is hyaluronan. Hyaluronan—which is a longer disaccharide polymer compared to chondroitin and heparan sulfate—is synthesized directly on the cell surface and anchored with a CD_44_ cell receptor [19]. Preventing hyaluronan loss has been suggested as particularly important in preventing the vascular complications of diabetes mellitus [6,20]. Dogné et al. [20] suggest that one therapeutic approach could be the inhibition of hyaluronidase 1, an enzyme responsible for cleaving hyaluronan from CD_44_ cell receptors. However, given that proteoglycans and glycosaminoglycans are not specific to the endothelial cells, but are ubiquitous in the human organism, the inhibition of cleavage enzymes might negatively affect functions in other cells. Singh et al. [21] found that immunoglobulin G (IgG) *N*-glycosylation patterns in type 2 diabetes were associated with a faster decline of kidney function, thus reflecting a pro-inflammatory state of the IgG. In that regard, studying the glycosylation patterns of eGC components might provide a more precise way of combating vascular symptoms.

To date, the main functions of the eGC layer in homeostatic conditions are protection of the endothelial cell membrane, mechanotransduction, the regulation of vasodilatation, and the prevention of blood clot and plaque formation (Figure 3) [22]. Another important function of this layer is the promotion of blood flow homogeneity, which may be a useful strategy for improving tissue perfusion in many [23]. Most eGC components, especially the glycosaminoglycans, have anionic-ending molecules such as uronic or salicylic acid, thereby giving the eGC a net negative charge in homeostatic conditions. The net negative charge allows the eGC to interact with blood proteins through ionic interactions or create repulsion against platelets and leukocytes (Figure 1 and Figure 2A) [22,24]. A healthy eGC may act as a sodium buffer by controlling the influx of sodium into the endothelial cells [25]. This was proposed because positively charged sodium is most likely to be bound to negatively charged organic material, which would provide for its osmotic inactivity. The ‘perfect’ sodium storage place would be the eGC, given its ubiquitous position and net negative charge in homeostatic conditions [25]. Related to this, eGC components, such as chondroitin sulfate, may have a therapeutic application in the targeted delivery of medical substances. This suggests an important role of eGC in the docking of oxidants, which may be the base for developing novel enzymatic antioxidant treatments [26]

In a healthy human body, the eGC is present in an extended state [27]. This means that the glycocalyx components exist in equilibrium with blood proteins, thus forming a thick and dense structure (see Figure 2A) [28]. The thickness and density of the eGC provide significant protection for the endothelial cells by preventing the platelets and leukocytes from adhering to the cell receptors on the cell surface. This is possible because the eGC is much wider, and the proteoglycans extend much further into the lumen of the blood vessel than the active sites of adherence receptors [29]. In addition, in homeostatic conditions, the rate of synthesis of adherence receptors is lower than in acute or chronic diseases, which further reduces the chance of blood component adherence [30] (Figure 2B).

The homeostatic eGC is often described as a vascular gatekeeper. Indeed, the firm and thick, but flexible and adjustable, structure can be imagined as a protective wall surrounding the castle, providing great protection against intruders, with the castle being the endothelial cell.

## 3. Extended eGC Activates Mechanotransduction and Endothelial Nitric Oxide Release

The so-called wind-in-the-trees conceptual model is sometimes used to describe one important function of the eGC: the mechanotransduction of blood-flow-induced fluid shear stress (Figure 4) [22]. Mechanotransduction is a process by which the eGC senses forces caused by fluid passage through the blood vessels and translates that information inside the endothelial cell to initiate intracellular signaling cascades. Upon receiving the signal, the endothelial cell acts in accordance with the given information by implementing, for example, structural maintenance, vasodilatation, senescence, or an inflammatory response [31]. This conceptual model describes the eGC components as the trees in a forest and the blood flow as the wind. The branches are the glycosaminoglycans, the tree trunks are the proteoglycan backbones, the cell receptors are the grass, and the endothelial cell membrane is the ground [22].

When the wind (fluid shear stress) blows through the forest, the treetops (glycosaminoglycans) sense it and move back and forth accordingly. The sensed force is then transduced onto the three trunks (proteoglycan core) and into the roots. The roots, in this case, would be the cytoskeletal structures and intracellular eGC domains. When the eGC is preserved (in the extended state), the winds’ impact (blood flow shear stress) is scattered between the branches of the treetops, whereas the ground senses little to none of the wind’s impact. However, the important information about the state of the blood flow is transduced via the trees into the roots. If the trees were not present or were sparse, the wind would directly interact with the ground and impact the ground, while the information could not be properly transmitted into the deeper structures.

Another theoretical review of the glycocalyx describes the mechanotransduction properties of the eGC by using a bumper car conceptual model. This model explains the processes inside the endothelial cell upon the arrival of signals from the blood. The bumper car analogy came to be after an experiment which showed that dense peripheral actin bands (DAPABs) were disrupted by uniform shear stress. In addition, the movement of vinculin, a cytoskeletal protein, closer to the cell membrane was detected. Both observations were enhanced when higher shear stress was implemented [32]. These observations suggest that cytoskeletal reorganization indeed occurs as a consequence of mechanical signal transduction of the eGC. The same effect of cytoskeletal reorganization inside the endothelial cell was not observed when the eGC was destroyed by proteolytic enzymes. Mechanotransduction is an important mechanism used to trigger different pathways in the endothelial cell. For example, eGC mechanotransduction initiates the process of nitric oxide (NO) production through the activation of endothelial nitric oxide synthase (eNOS) through calcium/calmodulin activation and consequential Ca^2+^ influx [31] (Figure 4). Endothelial-derived NO exerts powerful localized vasodilatory effects on the vascular smooth muscle cells [33]. Additionally, NO contributes to the inhibition of platelet aggregation and leukocyte adhesion. As NO is produced when eGC is preserved, the same functions can be contributed to a healthy eGC, and a proper NO production could be an indicator of eGC health.

## 4. The eGC in a State of Low-Grade Inflammation—Diminished eGC

### 4.1. eGC Response to Inflammation

The state of inflammation, led by the activation of the TNF-α pathway, triggers the collapse of the eGC (described in the chapter below). The collapse of the eGC is much needed in acute conditions such as injury or infection, as this promotes endothelial cell interaction with leukocytes/platelets and blood clot formation. However, certain eGC disruptors (described in detail later) can induce low-grade chronic inflammation, which may lead to the development of vascular consequences of chronic diseases if disruptors continuously trigger the shedding or endocytosis of eGC. Shedding and endocytosis are two main responses of eGC components to various disruptions from the environment. Both responses cause the collapse of the eGC and disrupt the dynamic equilibrium. For example, acting on an environmental cue, a family of enzymes collectively known as sheddases or secretases can be released and proteolytically cleave the proteoglycans’ ectodomain, leaving it virtually intact (Figure 2B, element 4). Little is known about the exact structure of the eGC sheddases. Novel findings by Yang et al. [34] revealed that a sheddase called disintegrin and metalloproteinase 15 (ADAM15) cleaved the CD_44_ surface anchor for hyaluronan. These kinds of shedded ectodomains travel via the bloodstream, where they may function as autocrine or paracrine effectors. The shedding mechanism both generates soluble ectodomains and rapidly reduces the amount of cell surface heparan sulfate [35]. Heparan sulfate reduction is much needed in conditions following acute injury, where platelet aggregation is needed for wound healing [36]. On the other hand, different ligands from the blood can bind to surface glycosaminoglycans and be internalized into the endothelial cell and serve as the cell’s nutrient source. Viruses can sometimes hijack this mechanism of ligand transport. For example, early-stage research has shown that the spike protein of SARS-CoV-2 interacts not only with angiotensin-converting enzyme 2 (ACE2), but with heparan sulfate as well, via the process of endocytosis or internalization [37]. These findings suggest that eGC knowledge might be important in formulating virus treatments, especially for the vascular consequences of the disease such as hypercoagulability and acute coronary symptoms.

### 4.2. Mechanisms of eGC Collapse

TNF-α and TNF signaling are hallmarks of inflammation and have been related to the cardiovascular pathophysiology of atherosclerosis, sepsis, diabetes, and obesity, among others [38]. TNF-α is a master regulator of proinflammatory mediators, and the activation of the TNF-α metabolic pathway has a major impact on the eGC structure [38]. Studies have demonstrated that acute exposure to TNF-α or thrombin, another inflammatory mediator, causes rapid shedding of the glycocalyx structure [16,39]. This process results in increased vascular permeability, tissue oedema, coronary leakage, and mast cell degranulation [40]. TNF-α activation has a negative impact on the eGC integrity through several mechanisms, e.g., by the induction of reactive oxygen and nitrogen species (ROS/RNS) [12] or by activating the NF-κB metabolic pathway [41]. Certain ROS/RNS can then cleave the ectodomains of eGC constituents via the activation of matrix metalloproteinases (MMPs) and inactivation of endogenous protease inhibitors [15]. Enhanced NF-κB signaling leads to the syndecan-4 domain synthesis [42]. In these inflammatory conditions, the synthesized syndecan-4 domains are rapidly shed from the endothelial cell surface. This shedding causes a disruption in the eGC integrity, but it is also proposed that, when present as a soluble molecule in the blood, syndecan-4 may facilitate tissue fibrosis [42]. SIRT-1-deficient endothelial cells have been shown to exhibit increased NF-κB pathways, thus shedding syndecan-4 ectodomains [43,44].

## 5. eGC in Chronic Diseases

Given that eGC destruction, due to environmental factors is inherent to several chronic pathologies, it is difficult to ascertain the contribution of eGC integrity to vascular disease progression/severity. As seen with metabolic syndrome, these kinds of diseases can be unified when constructing prevention tactics to preserve the eGC. The pathophysiology of important chronic conditions and the importance of eGC is highlighted in the following section.

### 5.1. Hypertension

As mentioned previously, one of the proposed eGC functions is to buffer plasma sodium and control its influx into the endothelial cells [25]. These findings are particularly important when studying hypertension. In one cross-sectional study, newly diagnosed hypertensive patients (*n* = 320) had decreased eGC thickness compared to healthy controls (*n* = 160). Reduced eGC thickness was related to signs of impaired vascular function, including increased central systolic blood pressure, as well as increased pulse wave velocity. These findings suggest that eGC thickness is reduced in untreated hypertension [5]. However, the question of the primary disruptor of eGC destabilization in hypertension is not yet definitively answered. Is it merely a consequence of mechanical destruction by increased blood pressure, or is it due to high salt intake, which may cause eGC collapse that has arterial stiffness and increased blood pressure as a consequence? The current evidence seems to suggest the latter. One study showed that a mere 2% increase in plasma sodium beyond 140 mM may stiffen the endothelial cells by approximately 20% [45]. In a related study, five days of sodium overload led to a ~50% eGC destabilization and 68% reduction of heparan sulfate [46]. It seems that, in the state of daily sodium excess, the eGC diminishes (Figure 2) and loses the sodium buffer capacity.

The first study to investigate the effect of salt (sodium chloride) on microvascular densities using in vivo sublingual imaging was performed by Rorije et al. [47]. The researchers aimed to determine if high salt intake would reduce sublingual microvascular density and, therefore, reduce eGC thickness in normotensive individuals (*n* = 18, all-male, mean age 29 ± 5 years). No blood pressure or sublingual microvascular differences were found when comparing high and low salt intake groups. However, an increase in salt consumption significantly correlated with a lower recruitment rate of sublingual capillaries after the administration of nitroglycerin, thus indicating lower structural microvascular density. These findings also suggest that a high salt load by itself might be one of the first disruptors of the eGC, which may lead to the cardiovascular damage observed in hypertensive patients. This study opens an array of questions and discussions worth exploring further in the attempt to prevent cardiovascular disease occurrence.

### 5.2. Atherosclerosis

Although research in humans is scarce, the role of the eGC in atherosclerosis in animal models is well recognized. A healthy, extended eGC prevents the occurrence of atherosclerotic plaque by ionically repelling macromolecules and mechanically preventing their adherence to the endothelial cell, thus lowering the chance for their migration and oxidation [4]. In animals, vascular sites with compromised eGC and lower mechanotransduction function seem to be more vulnerable to inflammatory consequences and the formation of atherosclerotic plaque [48,49]. The results from animal studies have led to an attractive hypothesis, which suggests that clinically assessing the thickness of the eGC might allow for the early detection of atherosclerosis. However, studies conducted in humans display contradicting results. For example, a clear connection between eGC thickness and vascular risk or progression/severity of vascular diseases has not been demonstrated. In one multi-ethnic cross-sectional study (*n* = 6169, 42.4% male, mean age 43.6 ± 13), the eGC thickness was assessed by using side-stream dark-field imagining, which is considered to be the most suited method according to some sources [50]. Reduced eGC thickness was associated with the female sex and diabetes after correcting for possible confounders such as age, diastolic blood pressure, and body mass index. Reduced eGC thickness was not associated with prevalent cardiovascular disease [51]. This study questions the viability of the used measurement technique as a proper measure of eGC thickness and the involvement of the eGC in atherosclerosis pathophysiology.

However, the cardiovascular biomarkers used in the study by Valerio et al. [51] (LDL cholesterol, HDL cholesterol, and triglycerides) might not show a full picture of the cardiovascular disease risk and, therefore, might not be reliable in comparing the consequences of reduced eGC thickness [52]. In a certain part of the population, namely, people with insulin resistance, the measures of LDL cholesterol can be in discordance with apolipoprotein B (apo B) and LDL particle concentration [53]. The number of apoB-containing lipoprotein particles is sometimes more predictive of high cardiovascular disease risk than the cholesterol content (LDL-C). In addition, the smaller size of the lipoproteins could be more indicative of future cardiovascular disease than the number of LDL-C packed in these molecules, due to their greater reactivity [54]. When assessing the connection between the eGC thickness and cardiovascular disease risk, other cardiovascular risk assessment parameters might be a good addition to the lipid panel.

### 5.3. Abdominal Obesity

The discordance in atherosclerosis risk assessment described in the chapter above is particularly important in people with diagnosed metabolic syndrome and type 2 diabetes [53]. Those conditions are usually accompanied by increased visceral and ectopic fat, which are metabolically different from subcutaneous fat and contribute to proinflammatory, proatherogenic, and procoagulant states [55]. Visceral fat in the abdomen and ectopic fat surrounding the liver, heart, kidneys, or pancreas accumulate as a consequence of a chronically positive energy balance. When a person reaches their so-called “fat threshold”, the subcutaneous fat loses its ability to expand through adipocyte hyperplasia. In that state, the body responds by increasing visceral fat deposits, thus increasing the production of proinflammatory cytokines such as TNF-α and interleukin 6 [55]. In addition, ectopic fat deposits can increase the hepatic production of glucose, which is a process linked to glucose intolerance. The increased TNF-α activity leads to an eGC degradation by activating ROS/RNS and NF-κB metabolic pathway [55]. Furthermore, increased glucose production can lead to AGE formation and further eGC destruction (see ‘The effects of elevated blood glucose on eGC’).

### 5.4. Hyperglycemia, Type 2 Diabetes, and Metabolic Syndrome

Chronically increased blood glucose (i.e., hyperglycemia), a hallmark of diabetes, metabolic syndrome, and obesity, causes the increased production of advanced glycation end-products (AGE) [56]. AGEs are glycated proteins, lipids, and nucleic acids commonly present in homeostatic conditions but can be rapidly generated in pathological conditions, such as insulin resistance [56]. Generated AGEs upregulate the iNOS system in chronic uncontrolled hyperglycemia. When upregulated, the iNOS system causes eNOS dysfunction and contributes to the inflammatory conditions of the body [57]. Both eNOS and iNOS systems used l-arginine and molecular oxygen to produce l-citrulline and NO. However, when NO is produced in inflammatory conditions, it quickly reacts with superoxide (O_2_^−^) and generates peroxynitrite (ONOO^−^), which causes nitrosative stress and damages proteins, lipids, and DNA [58]. When activated, iNOS further enhances the generation of oxidative stress, which is a powerful process in eliminating microbial infections and tumor cells but may also contribute to chronic disease development [59,60]. Reactions between O_2_^−^ and NO lower NO bioavailability, which can cause arterial stiffness and promote atherosclerotic events [61]. AGEs form a cross-link between the basement membrane molecules of the extracellular matrix [62], thus implying their close relationship with the eGC; however, the interaction between AGEs and eGC, as well as the implications for chronic disease development, are largely unknown.

AGEs activate various intercellular signals through receptor- and non-receptor-mediated mechanisms. For example, receptors for advanced glycation end-products (RAGEs) may, in part, explain the relationship between the eGC and AGEs. Accumulated AGEs activate RAGE receptors that have been shown to be present on the endothelial cells and can be shed from the surface in a manner similar to eGC components, thus forming soluble RAGEs (sRAGEs) [63]. sRAGEs upregulate the NF-κB pathway and facilitate the inflammatory cascade through the release of ROS/RNS. ROS/RNS contribute to the shedding of eGC components and enhance endothelial dysfunction. It has been shown that AGE-bound RAGEs increase endothelial permeability to macromolecules and block NO signaling activity [64]. This is perhaps explained by the destruction of the eGC (i.e., the removal of the endothelial gatekeeper), thus promoting the accessibility of blood component migration to the endothelial cell.

A growing body of evidence finds that AGEs can derive from food sources and tobacco, and they suggest dietary AGE restriction [65,66]. In addition to AGE pro-oxidative effects, hyperglycaemia might also be responsible for altering the sulfation patterns of glycosaminoglycan chains and can prevent hyaluronan binding to the glycocalyx [40]. We have shown that 14 days of sucrose supplementation (3 × 75 g of sucrose per day) impairs vascular function in young healthy male subjects that was indicated by blood flow during passive leg movement, which is a method suited for determining the impacts on mechanotransduction in vivo [67]. The high sucrose ingestion affected the vasodilatory properties of the vessels, reduced eNOS activation, and upregulated PECAM-1. Given that a healthy eGC is important for proper mechanotransduction and that PECAM-1 is upregulated in oxidative stress that might have been caused by increased sucrose intake, these findings suggest a disruption of the eGC. Future research is warranted to investigate the influence of dietary sugar on the eGC, particularly regarding eGC shedding and integrity.

The effect of accumulated abdominal fat is further augmented by chronically elevated glucose levels, which can cause insulin resistance, which is a hallmark of metabolic syndrome, diabetes, and prediabetes [68]. In addition, high postprandial glucose levels generate AGEs, which further activate proinflammatory signaling pathways [66] (see ‘The effects of elevated blood glucose on eGC’). Pertynska-Marczewska and Merhi [69] researched the role of the AGE–RAGE axis in the prevention of atherosclerosis in women in menopause. They found that circulating sRAGE levels could be correlated with increased abdominal fat, insulin resistance, diabetes, and metabolic syndrome. They suggested that a therapeutic inhibition of the RAGE signaling pathway might be beneficial for decreasing cardiovascular disease risk in women in menopause [69]. Indeed, as discussed previously, Valerio et al. [51] reported a correlation between diabetes and low eGC thickness, thus implying “a small glycocalyx size in people with diabetes” (i.e., the diminished/collapsed state of the eGC). This is in accordance with other research that showed that both acute and chronic hyperglycaemia significantly reduced eGC size, particularly in patients with microalbuminuria [70]. Both acute and chronic effects of hyperglycaemia on vasculature have been recorded and are evident from the fact that the comorbidities of type 2 diabetes are closely related to the degradation of the vascular system and eGC health [70].

### 5.5. Chronic Kidney Disease

One of the adverse consequences of diabetes is diabetic nephropathy, which is, next to hypertension, a major cause of end-stage renal disease [71]. The role of the eGC in the pathophysiology of diabetic nephropathy is well recognized, and efforts have been currently made to produce specific therapies that target the regeneration of the glycocalyx of the fenestrated glomerular endothelial cells and the prevention of albuminuria [72]. Albuminuria is a pathologic state of increased urine albumin due to improper glomerular filtration partially caused by the destruction of the glomerular endothelial cell’s glycocalyx [72]. It has been shown that people with albuminuria caused by diabetic nephropathy have drastically increased levels of heparinase and hyaluronidase, which cause shedding of the glycocalyx [73]. One of such efforts is the inhibition of monocyte chemotactic protein-1 (MCP-1), which is a protein that activates the migration of inflammatory cells such as monocytes and macrophages to the kidney. Those infiltrated glomerular macrophages can secrete cathepsin L, which is proposed to be responsible for heparinase activation [74]. Boels et al. [75] showed that MCP- 1 inhibition significantly reduced albuminuria in diabetic nephropathy and restored glomerular eGC dimensions.

### 5.6. Chronic Inflammation

Lipopolysaccharides or endotoxins are components of the outer membrane of Gram-negative bacteria and are often used to trigger inflammation in experimental studies. In humans, lipopolysaccharides usually originate from the skin, local infections, and mucosal membranes. In some instances, lipopolysaccharides may cause endotoxemia, which is marked by the activation of TNF-α and iNOS inflammatory pathways and a consequential increase in oxidative stress and inflammation [76]. Inagawa et al. [77] noticed that the eGC of the lungs was severely diminished under experimental endotoxemia conditions. These findings suggest a causal relationship between the disruption of the eGC and microvascular endothelial dysfunction, which is a characteristic of sepsis-induced acute respiratory distress syndrome. In one pioneer research, Li et al. [78] showed that 100 ng of maresin conjugates in tissue regeneration 1 (MCTR1) increased the survival rates of mice from lipopolysaccharide-induced sepsis. These researchers also found a reduction in serum heparan sulfate and syndecan-1 levels in mice treated with MCTR1 compared to the control group, which indicates lower rates of eGC shedding. MCTR1 is produced in macrophages by the 14-lipoxygenation of docosahexaenoic acid (DHA) [79]. This fact may show importance in researching the connection between the eGC and the dietary intake of DHA (discussed later). eGC destruction might be connected to chronic inflammatory bowel diseases (IBDs) such as Crohn’s disease and ulcerative colitis. IBDs are marked by chronic low-grade inflammation, which is believed to originate from the gut endotoxins [80]. People with IBD have higher rates of intestinal permeability and serum levels of lipopolysaccharides, which may negatively influence the eGC by activating the TNF-α inflammatory pathway, which destroys the eGC integrity [81].

Sodium glucose co transporter 2 (SGLT2) inhibitors are administered in the treatment of both type 2 diabetes and chronic kidney disease, and the positive effect of SLGT2 inhibitors on vascular function may be related to eGC recovery. Decreased oxidative stress seem to be one of the mediators of the effects of SGLT-2 inhibitors, and since ROS are known disruptors of the endothelial glycocalyx, SGLT2 inhibitors may improve mechanotransduction, restore nitric oxide production, and improve vasodilation. Future research should aim to confirm this hypothesis.

## 6. Perspectives of Nutritional Therapy for eGC Health

The presented evidence suggests a close connection between the pathophysiology of various chronic diseases and points towards eGC destruction as a driver of endothelial dysfunction and consequent vascular injury. Any chronic damage to the eGC can result in vascular permeability, oedema, platelet aggregation, and aprothrombotic environment, which are consequences that are well recognized in the end stages of chronic conditions such as diabetes, hypertension, metabolic syndrome, and obesity [40]. Defining the eGC disruptors and regenerative compounds might be the future of eGC prevention or recovery therapy in both healthy and diseased individuals. A few potential nutritional and behavior-related therapies are discussed in the chapter below.

### 6.1. Preventing Vitamin D Deficiency

The first described functions of vitamin D were related to immunity, viral disease, and autoimmune disease prevention [82]. This ancient function is being rediscovered, with findings suggesting an association between vitamin D deficiency and COVID-19 symptoms, particularly thrombosis and coagulopathy, which are the same symptoms related to heparan sulfate loss [83,84]. Given that heparan sulfate is one of the SARS-CoV-2 receptors, a logical postulate would be that improper heparan sulfate synthesis in COVID-19 is responsible for some of the observed symptoms. In recent times, vitamin D has also been connected to the preservation of the endothelial function through monocyte adhesion prevention and inflammation reduction [85]. This suggests that vitamin D could be an essential element for preserving or regenerating the eGC. Future research studies are warranted to investigate the role of vitamin D on eGC health and patient groups that will administer vitamin D as therapy for, e.g., hyperparathyroidism or osteoporosis, which might provide the first line of evidence.

### 6.2. Vitamin D and eGC Connection Hypothesis

For vitamin D to be activated, it needs to be converted into the hormonal 1,25-OH vitamin D. The enzymes needed for that conversion are the vitamin D receptor (VDR) and 1-a-hydroxylase, both of which can be found in cardiovascular tissues. Particularly, the VDR is expressed on the endothelial cells and is upregulated under stress [86]. When activated, the VDR affects calcium influx across the endothelial cell membrane, which is needed for eNOS activation and proper nitric oxide release. As mechanotransduction signals arising from the eGC activate this process, a close interaction between vitamin D and eGC has been suggested [87].

Vitamin D has a role in immunity; particularly, it has been shown that vitamin D regulates apoptosis and autophagy. One of its protective mechanisms is thought to be the inhibition of superoxide anion generation, NF-κB, and TNF-α [88]. In one randomized controlled trial, Omidian et al. [89] found a significant reduction in TNF-α levels when supplementing diabetic patients with 4000 IU/day of vitamin D for three months. This evidence is another argument in favor of the protective role of vitamin D on eGC, as TNF-α degrades the glycocalyx.

### 6.3. Supplementing with Omega-3 Fatty Acids and Probiotics

Combined supplementation with probiotics and omega-3 (Ω-3) fatty acids might be important in eGC regeneration. Probiotic strains such as Lactobacilli and Bifidobacteria lower lipopolysaccharide-dependent chronic low-grade inflammation by inhibiting the binding of lipopolysaccharide to the CD_14_ receptor, thereby reducing the overall activation of NF-κβ. Ω-3 fatty acids have been shown to increase Bifidobacteria—via unclarified mechanisms—which then suppress lipopolysaccharide and decrease lipopolysaccharide-producing bacteria, such as Enterobacteria [90]. Supplementation with Ω-3 has repeatedly been shown to decrease endothelial dysfunction and increase vasodilation and vessel elasticity, as well as decrease inflammatory pathways [91,92,93].

Taken together with the proposed vitamin D connection, the presented evidence suggests that interactions between lipid metabolism and eGC might be particularly relevant to research further. When conducting such studies, the type of dietary fat is an important factor to consider. Some studies suggest that fat-rich and energy-rich diets are the main source of increased endotoxemia, whereas unsaturated fatty acids have been associated with lower postprandial circulating levels of lipopolysaccharides [90,94,95]. Caution must be taken with the potential cofounders. Dietary sugars and salt might also be eGC disruptors, so their influence should be considered when forming further hypotheses and experiments.

### 6.4. Providing the eGC Building Blocks

Currently, there are two nutraceuticals on the market that have been developed with the purpose of eGC regeneration. Various other therapies such as metformin, rosuvastatin, hydrocortisone, sulodexide, and heparin have been proposed as having eGC regenerative properties [96]. The main premises of the developing therapies are to provide the eGC with the building blocks for quicker regeneration or to remove eGC disruptors. An important animal study showed that a 10-week treatment that targeted the eGC by using high molecular weight hyaluronan and other eGC components improved eGC properties and ameliorated age-related arterial dysfunction in old mice. The findings suggest that the eGC may be a potential therapeutic target for treating age-related arterial dysfunction [97]. 

Healthy dietary patterns, particularly the Mediterranean diet (MedDiet), Nordic diet, Traditional Asian diet, and Dietary Approaches to Stop Hypertension (DASH), have been shown to be beneficial in reducing the risk of arterial dysfunction and other diet-related chronic diseases [98,99,100]. Compared to a typical Western-style diet, these kinds of dietary patterns are characterized by lower trans fat and lower excess sodium and sugar consumption (lower meat and processed food intake), higher fiber intake (from whole grains and legumes), higher fruit and vegetables content, and higher Ω-3 content (from fish and nuts) [101]. Studies suggest that the MedDiet is suitable as a type 2 diabetes therapy, as it was associated with improved glycemic control when compared to a control dietary pattern [102]. In addition, some long-term randomized controlled trials and meta-analyses showed a greater chance of remission from metabolic syndrome following the MedDiet and a significant reduction in stroke incidence [102,103]. One possible explanation for the observed health benefits could be that healthy dietary patterns contain the building blocks for preventing the pathological destruction and/or regenerating the eGC, thus decreasing cardiovascular disease risk. Further large-scale studies are needed to confirm the connection to healthy dietary patterns, which might also be useful as a source of the eGC building blocks and, therefore, may decrease cardiovascular disease risk. Further large-scale studies are needed to confirm the connection [104]. Other potential beneficial dietary components are discussed in the chapters that follow.

### 6.5. Dietary Sulfur

Sulfur-containing compounds might help reduce eGC damage by exhibiting antioxidative and anti-inflammatory properties and are, therefore, good candidates to be implemented in preventative nutritional therapy [105]. High levels of sulfur can be found in meat and fish, as sulfur is a part of the sulfur-containing amino acids (methionine and cysteine). However, Doleman et al. [106] state that an impressive 89.5% of dietary sulfur in a typical diet derives from other sources due to the differences in the distribution of types of food. Intake from alliaceous and cruciferous vegetables contributed to almost half of the total sulfur intake. Both alliaceous vegetables (for example onion, garlic, or leek) and cruciferous vegetables (such as broccoli, kale, asparagus, or mangold) are abundant in various healthy dietary patterns. In alliaceous vegetables, sulfur is a part of organosulfur compounds and is known for its effectiveness in eliminating viral and bacterial infections [107]. Interestingly, one double-blinded placebo-controlled randomized study found that aged garlic extract may protect and slightly improve the microcirculation in patients with a Framingham Risk Score ≥ 10 (increased risk for cardiovascular disease) [108]. To date, this is the largest study researching the effects of garlic extract on microcirculation. 

Sulfur is also a structural part of isothiocyanates, which are, in a broad range, found in cruciferous vegetables, which are staples of healthy dietary patterns [105]. Sulforaphane, the most researched isothiocyanate, exhibits anti-inflammatory and antioxidative properties that have been confirmed in various in vivo and epidemiological studies and may reduce levels of fasting blood glucose and glycated hemoglobin [109], as well as AGE concentration [110]. When researching the effects of sulforaphane in mice models of skin cancer, Alyoussef and Taha [111] found that sulforaphane blocked sulfatase-2 activity which, when activated, significantly elevated heparan sulfate proteoglycan concentration in plasma. The main function of sulforaphane is thought to be the activation of the antioxidative nuclear factor E2-related factor 2 (Nrf2) metabolically pathway, which ameliorates excess oxidative stress upon activating numerous cytoprotective proteins [111]. One of those proteins is metallothionein, a cysteine protein that binds copper and zinc as cofactors. When activated, metallothionein can extinguish ROS/RNS due to its high thiol content [112]. In animal models of type 1 diabetes, sulforaphane and zinc have proven to be more cardioprotective when combined [112]. Human studies are needed to further substantiate this evidence. The eGC is a ubiquitous structure with a relatively high amount of sulfur-containing components such as heparan and chondroitin sulfates, which are in an almost constant state of metabolic turnover [17]. Aside from its antioxidative properties, dietary sulfur might be an important sulfur donor and could provide the building blocks for eGC regeneration.

### 6.6. Dietary Nitrates

Dietary nitrates are high in green leafy vegetables and some root vegetables, such as beetroot. In research, dietary nitrates are usually provided as beetroot juice or sodium nitrate and have been shown to reduce inflammation and thrombosis; however, the findings are not conclusive [113]. The explanation behind the beetroot juice intervention is based on the hypothesis that providing an exogenous source of nitric oxide might improve endothelial function. Most of the acute and short-term research on hypertensive patients in this area shows substantial improvements in resting blood pressure and muscle microvascular function, as well as reduced arterial stiffness [114,115]. On the contrary, longer (7 days) but similarly designed studies did not find improvements in cardiac function or endothelial integrity in healthy non-smoking adults [116]. Furthermore, two-week supplementation of beetroot juice was insufficient to improve blood pressure or endothelial function in type 2 diabetics [117].

One of the possible explanations for the discordance in evidence might be the fact that a healthy and extended eGC is responsible for proper nitric oxide production [31]. Providing the end product (nitric oxide) will not regenerate the eGC but can be beneficial in providing nitric oxide in, for example, people who are newly diagnosed with a chronic vascular disease, based on the observation that eGC is destroyed in these conditions. In that case, beetroot juice provides a secondary source of nitric oxide and acutely helps in the preservation of vascular function until the eGC is fully restored and regains the ability to produce endogenous nitric oxide. This hypothesis would explain the acute beneficial effect and long-term stagnation of the results seen in some studies and in healthy subjects.

### 6.7. Lifestyle Changes

Dietary and lifestyle modifications that promote weight loss in overweight and obese individuals have been shown to be beneficial in decreasing vascular fat storage and inflammatory molecule production [118]. In recent years, professional-led weight loss even helped in completely reversing type 2 diabetes by impacting secondary insulin resistance to reduced hyperinsulinemia [119]. In that sense, weight loss could be a powerful tool in preventing eGC damage, as it combats many eGC disruptors simultaneously. Collectively, weight loss can eliminate inflammatory cytokines generated by visceral fat, decrease blood glucose levels, and prevent the formation of AGEs [54]. The combination of a healthy dietary pattern, caloric restriction, and physical activity has been used in clinical settings and shows promising results for treating non-alcoholic fatty liver diseases, which are frequently present in patients diagnosed with diabetes, insulin resistance, and obesity [120].

Intermittent fasting, prolonged fasting, time-restricted eating, and similar dietary strategies are based on restricting daily time defined for eating and prolonging fasting time [121]. The beneficial effect of intermittent fasting on vascular health parameters, microcirculation, and vasodilatation has been detected, even in the absence of weight loss in both healthy individuals and men with prediabetes [122,123]. The observed effects were related to lower blood pressure, increased insulin sensitivity, decreased oxidative stress, and higher levels of nitric oxide release. However, the mechanisms of the positive effects are not yet fully discovered [124].

The eGC might hold the key in explaining the observed positive effects of time-restricted dietary regimes; however, repeated uniform intermittent fasting studies with a reliable measure for eGC thickness are needed to confirm this hypothesis. If confirmed, intermittent fasting (or other interventions based on time restriction of eating) might be useful in nutritional therapy for vascular consequences of chronic diseases. Recent findings in this area suggest that setting the eating window earlier in the day may be optimal due to the natural circadian rhythms of humans which control hormone release, thus influencing stress levels and insulin resistance [125,126].

## 7. Conclusions

Deep and precise knowledge regarding the eGC seems to be the missing link in finding novel treatments for lifestyle-related diseases such as atherosclerosis, type 2 diabetes, hypertension, obesity, and metabolic syndrome. Changes in the eGC integrity might be an early sign of cardiovascular disease development. The detection of these changes could particularly be important for raising adherence to healthy dietary patterns and lifestyle interventions that have proven to be cardio-protective. Studying vitamin D, omega-3 fatty acids, probiotics, dietary sulfurs, and dietary nitrates could be promising in exploring the connection between diet components and eGC regeneration. Lifestyle interventions such as weight loss and time management of eating might be important for eGC regeneration. Increasing research in this area could provide more precise guidelines in chronic cardiovascular disease prevention.

## Figures and Tables

**Figure 1 nutrients-15-02573-f001:**
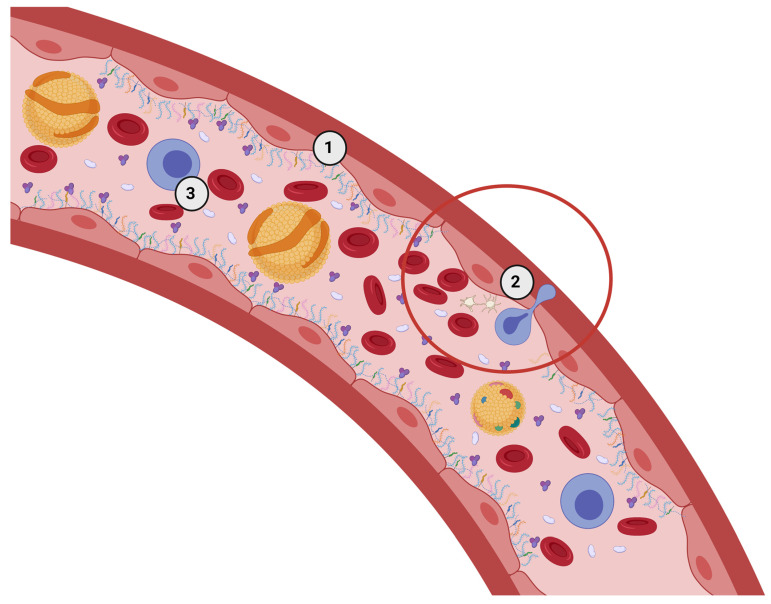
A model of the position of endothelial glycocalyx, which covers the apical surface of the endothelial cells (1). The circled area presents a part of the vessel without the glycocalyx (2). In this area, leukocytes, platelets, and other blood components can reach the endothelial cell receptors more easily and start blood clot formation, leukocyte migration, and other processes. This quicker adhesion is important in wound healing but might contribute to chronic disease progression as well. (3) Various blood components passing through a blood vessel. Created with BioRender.com.

**Figure 2 nutrients-15-02573-f002:**
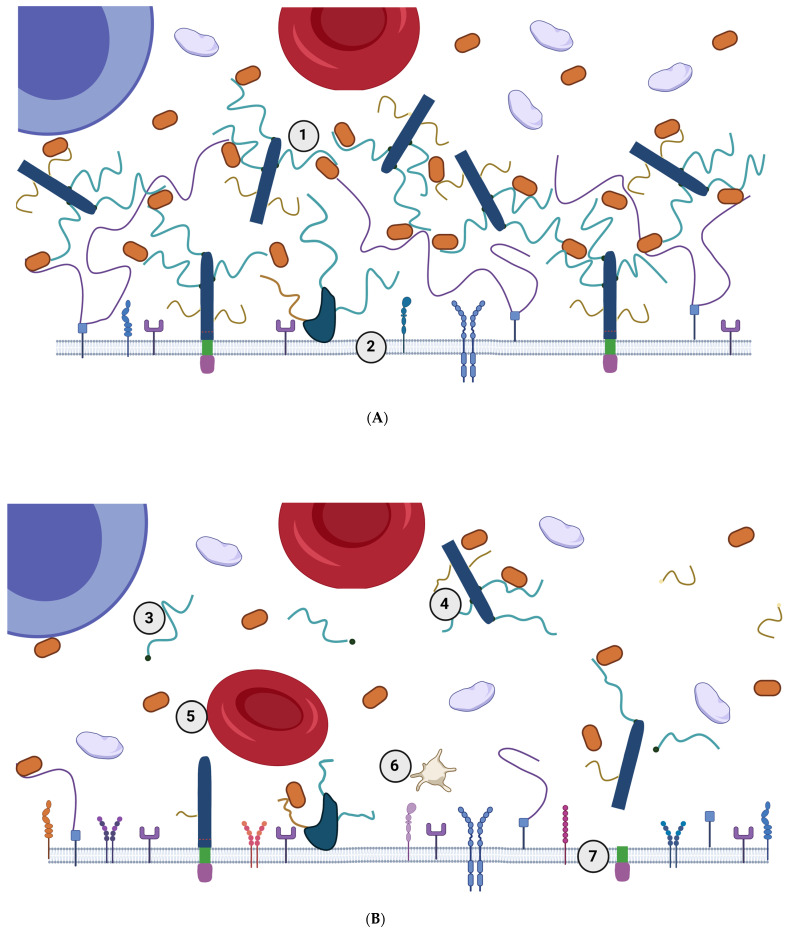
(**A**) The extended state of the endothelial glycocalyx. Due to their negative charge, the glycocalyx components are in dynamic equilibrium with positively charged plasma proteins, which are depicted as orange ellipses (1). Some of the components are not membrane-bound but contribute to the thickness of the whole structure. The thickness and the net negative charge prevent the blood components from adhering to the cell receptors. The cell membrane (2) does not contain many adherence receptors. (**B**) The collapsed state of the endothelial glycocalyx. Different disruptors can cause eGC to diminish or collapse, either directly or by triggering proteolytic enzymes that cleave different structures, which, therefore, destroy the equilibrium with plasma proteins. Those kinds of proteolytic enzymes are usually upregulated in inflammation. The eGC structures detach from the cell surface and become bloodborne. They may be measured in the blood. The high plasma concentration of, for example, heparan sulfate (3) or syndecan-1 (4) indicates eGC injury. In inflammation, blood components, such as red blood cells (5) or platelets (6), have easier access to the cell surface, and the expression of the adherence receptors on the cell membrane is upregulated (7). eGC–endothelial glycocalyx.

**Figure 3 nutrients-15-02573-f003:**
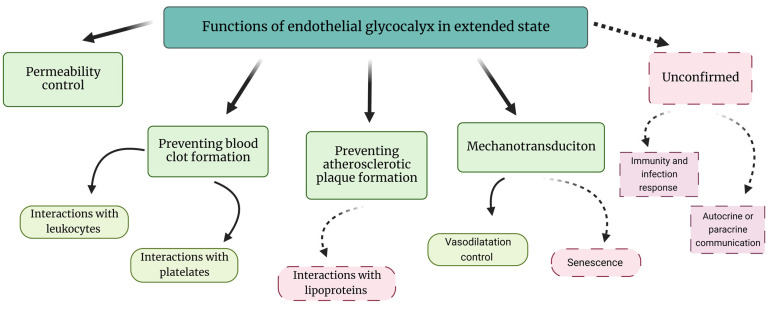
Functions of the extended state of the endothelial glycocalyx. The functions in green boxes have been studied in more depth. The functions in red and dashed boxes are currently not well understood. Solid lines are established connections, dashed lines are potential connections. In an eGC diminished/collapsed state, the functions of the endothelial glycocalyx are practically reversed. The collapsed state allows blood clot formation and plaque formation, while mechanotransduction function is reduced. Created with BioRender.com.

**Figure 4 nutrients-15-02573-f004:**
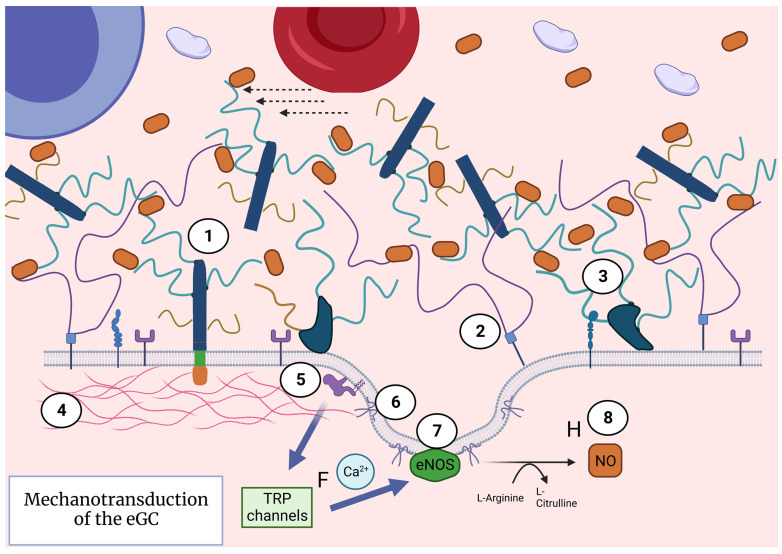
The mechanism of mechanotransduction of a healthy, extended eGC. The syndecans (1), hyaluronans (2), and other glycoconjugates (3) sense the changes in fluid shear force. This mechanical signal is transduced inside the endothelial cell and causes the movement of cytoskeletal structures (4). Those cytoskeletal structures activate the G-protein (5), which activates the transient receptor potential (TRP) channels and causes Ca^2+^ influx (6). Ca^2+^ ions are needed to activate caveolin proteins on the cell membrane (7), which attach to the endothelial nitric oxide synthase (eNOS) system and activate it. Activated eNOS generates nitric oxide (NO) (8). NO can then diffuse into the lumen of the blood vessel or the smooth muscles and cause vasodilatation. Created with BioRender.com.

## Data Availability

No new data were created.

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
