# Peer review of "Endothelial Glycocalyx Preservation—Impact of Nutrition and Lifestyle"

_nutrients, 2023, doi:10.3390/nu15112573_

Round 1

Reviewer 1 Report

1. Can altered eGC interact with fibrin chains due to altered glycosylation and thus contribute to atherosclerotic lesions and complicated plaques in the development of adverse cardiovascular events?

2. Is there any significance of SGLT2 administration in glycocalyx repair considering that these are drugs for both t2DM and chronic renal failure?

3. Considering the positive effect of vitamin D on the preservation of eGC, are there any data on patients who are on regular vitamin D therapy in primary or secondary hyperparathyroidism or osteoporosis?

Author Response

Dear Reviewer,

Thank you for taking the time to review our manuscript and for providing valuable feedback. We appreciate the opportunity to revise our work in response to your comments. We have carefully considered each of your suggestions and have made the following replies:

Comment 1: Can altered eGC interact with fibrin chains due to altered glycosylation and thus contribute to atherosclerotic lesions and complicated plaques in the development of adverse cardiovascular events?

Response: There are not many studies that specifically investigated the interaction between the endothelial glycocalyx and fibrin chains. It is likely possible that altered endothelial glycocalyx can interact with fibrin chains due to altered glycosylation and contribute to the development of atherosclerotic lesions and complicated plaques, which can ultimately lead to adverse cardiovascular events. A likely hypothetical mechanism would be that altered glycosylation of the endothelial glycocalyx leads to increased fibrin deposition in the arterial wall. This, in turn, can contribute to the formation of atherosclerotic lesions and complicated plaques. More studies are needed to confirm.

Comment 2: Is there any significance of SGLT2 administration in glycocalyx repair considering that these are drugs for both t2DM and chronic renal failure?

Response: No such studies have been conducted but a few key findings may be important in forming a hypothesis that administration of SGLT2 inhibitors could lead to endothelial glycocalyx recovery. Improved vascular function and decreased oxidative stress seem to be the main beneficial effects of SGLT-2 inhibitors. Oxidative stress is known to lead to cell and tissue damage via the production of ROS, known disruptors of the endothelial glycocalyx. By inhibiting ROS production, SGLT2 inhibitors may improve mechanotransduction (a process partially activated by a healthy eGC), which leads to restored nitric oxide production, and improved vasodilation. We have included this discussion in the manuscript line 433.

Comment 3: Considering the positive effect of vitamin D on the preservation of eGC, are there any data on patients who are on regular vitamin D therapy in primary or secondary hyperparathyroidism or osteoporosis?

Response: These kinds of studies are not known to the authors and is highly unlikely such were ever conducted. The main barrier to conducting such studies is a reliable eGC detection technique. Measuring the components of the eGC in the blood has only proven to be useful in acute conditions such as sepsis. We propose that measuring the thickness of the eGC layer in vivo may be a more reliable parameter of eGC health in chronic conditions such as osteoporosis. Further development of such techniques by implementing machine learning-based software will make it easier to implement such measurements in experimental studies. We have included this discussion in the manuscript line 462.

We appreciate your careful review and your thoughtful comments, and we believe that your feedback has helped to make this a stronger paper. Please do not hesitate to contact us if you have any further questions or concerns.

Sincerely,

Paula & Lasse

Reviewer 2 Report

In the present manuscript, the authors aimed to summarize the current literature on the relationship between endothelial glycocalyx (eGC) preservation and the impact of nutrition and lifestyle in the face of chronic diseases. The author did a good job illustrating the basic structure and functions of eGC from a simplicity standpoint. There was also good amount of details on the mechanisms of how eGC plays a role in vascular function in healthy state as well as the impact of disruptors to eGC integrity. However, there was a poor progression from the impact of the individual disruptor of eGC integrity to the impact of chronic diseases on eGC which makes it seem redundant. Authors also barely mention any vascular function measurements in the “eGC in chronic diseases” section which is the critical link between eGC and the progression of theses chronic diseases. For the “perspectives of nutritional therapy for eGC health” section, study on Endocalyx, a high molecular weight hyaluronan supplementation, should be discussed (PMID: 36787090). For the “Lifestyle changes” section, it was a hasty end that authors did not provide sufficient level of details on how these lifestyle strategies can potentially rescue the eGC in chronic disease status and eventually ameliorate the chronic disease-related vascular function. There are several major questions that are not addressed that are listed in the comments below.

Major Comments:

1.    Regarding the functions of glycocalyx, the manuscript is missing some important aspects such as its promotion of blood flow homogeneity (PMID: 27199117) and docking of oxidants (PMID: 22690296), cytokines (PMID: 14704229).

2.    Excess salt intake leads to elevated systolic blood pressure and augmented arterial stiffness in mice (PMID: 36735405), which seems like arterial stiffness precedes hypertension, should be discussed in the manuscript. The same mice model also shows a decreased glycocalyx thickness and increased permeability, which could be the underlying reason for the deteriorated arterial function (PMID: 34995168).

3.    Authors need to define which diameter category of the vessels were they focusing on for the eGC.

4.    Authors should consider using “diminished eGC” or to replace for “collapsed eGC” as the latter has the implication of a totally squashed glycocalyx which is misleading to the readers. Moreover, a collapsed eGC could be considered transient.

5.    Figures 2 and 4 were difficult to understand because it was not clear what shape the number corresponds to.

Minor Comments:

1.    Line 76, “given that that” changes to “given that”

2.    Line 185, “EGC’ changes “eGC”

Author Response

Dear Reviewer,

Thank you for taking the time to review our manuscript and for providing valuable feedback. We appreciate the opportunity to revise our work in response to your comments. We have carefully considered each of your suggestions and have made the following replies:

Comment 1: There was a poor progression from the impact of the individual disruptor of eGC integrity to the impact of chronic diseases on eGC which makes it seem redundant. Authors also barely mention any vascular function measurements in the “eGC in chronic diseases” section which is the critical link between eGC and the progression of these chronic diseases.

Response: Given the very few human studies of eGC and various chronic conditions, it is dificult to make a solid progression from indications to causal links between eGC and progression of chronic diseases. Our intention was to highlight the potential in this area and thus try to but all bits and pieces into this. We have now merged the two sections in an effort to make the connection between individual disruptors of eGC and chronic disease more clear.

Comment 2: For the “perspectives of nutritional therapy for eGC health” section, a study on Endocalyx, a high molecular weight hyaluronan supplementation, should be discussed (PMID: 36787090).
Response: A section about the current therapeutics and nutraceuticals for eGC regeneration have been added to the said section and the proposed study is discussed. At the time of writing, Endocalyx benefits were not yet confirmed to have an effect so we are delighted to see new evidence coming out in this burning area and are grateful that you pointed us in this direction.

Comment 3: For the “Lifestyle changes” section, it was a hasty end that authors did not provide sufficient level of details on how these lifestyle strategies can potentially rescue the eGC in chronic disease status and eventually ameliorate the chronic disease-related vascular function.

Response: The lifestyle changes is part of the perspectives section and we did not put too may speculative details in there. There is a lack of studies investigating eGC in relation to various lifestyle interventions but we are confident that such studies will soon provide evidence for a link. We have chosen to leave the „hasty“ paragraph in the manuscript to highlight this, but it can be omitted if the reviewer suggest this.

Comment 4: Regarding the functions of glycocalyx, the manuscript is missing some important aspects such as its promotion of blood flow homogeneity (PMID: 27199117) and docking of oxidants (PMID: 22690296), cytokines (PMID: 14704229).

Response: Thank you very much, there are now included in the review in the section about the eGC in healthy conditions.  

Comment 5: Excess salt intake leads to elevated systolic blood pressure and augmented arterial stiffness in mice (PMID: 36735405), which seems like arterial stiffness precedes hypertension, should be discussed in the manuscript. The same mice model also shows a decreased glycocalyx thickness and increased permeability, which could be the underlying reason for the deteriorated arterial function (PMID: 34995168).

Response: Thank you, this is well taken and we include this in the section hypertension.

Comment 6: Authors need to define which diameter category of the vessels were they focusing on for the eGC.

Response: We have defined this as the microcirculation in the introductionas most of the blood flow and pressure regulation occurs that the arteriolar level.

Comment 7: Authors should consider using “diminished eGC” or to replace for “collapsed eGC” as the latter has the implication of a totally squashed glycocalyx which is misleading to the readers. Moreover, a collapsed eGC could be considered transient.

Response: We have included “diminished eGC” instead of only collapsed throughout the manuscript.  

Comment 8:  Figures 2 and 4 were difficult to understand because it was not clear what shape the number corresponds to.

Response: The figure legends provide number along with a description of what the individual shape represents. We hope this helps the reader in understanding the story of each figure.

Minor Comments were all addressed.

  1. Line 76, “given that that” changes to “given that” - changed
  2. Line 185, “EGC’ changes “eGC” – changed

We appreciate your careful review and your thoughtful comments, and we believe that your feedback has helped to make this a stronger paper. Please do not hesitate to contact us if you have any further questions or concerns.

Sincerely,

Paula & Lasse